# Aptamers as Insights for Targeting SARS-CoV-2

**Suna Karadeniz Saygılı** [1,2], **Anna Szymanowska** [1], **Gabriel Lopez-Berestein** [1,3], **Cristian Rodriguez-Aguayo** [1,3] and **Paola Amero** [1,*]

1. Department of Experimental Therapeutics, The University of Texas MD Anderson Cancer Center, Houston, TX 77030, USA
2. Department of Histology and Embryology, Kutahya Health Sciences University, Kutahya 43100, Turkey
3. Department of Cancer Biology, Center for RNA Interference and Non-Coding RNA, The University of Texas MD Anderson Cancer Center, Houston, TX 77030, USA
* Correspondence: pamero@mdanderson.org; Tel.: +1-713-563-3993

**Abstract:** The Severe Acute Respiratory Syndrome coronavirus (SARS-CoV-2) continues to be a major cause of high mortality in the world. Despite many therapeutic approaches having been successfully developed, there is still the need to find novel and more effective therapeutic strategies to face the upcoming variants. Here, we will describe the potential use of aptamers, synthetic single-stranded oligonucleotides, as promising tools to target SARS-CoV-2. Since aptamers have been successfully developed against viruses, this review will focus on the latest selection approach method using artificial intelligence, the state-of-the-art in bioinformatics, and we will also summarize the latest discoveries in terms of aptamers against spike protein and other novel receptor proteins involved in SARS-CoV-2 entry and the use of single-cell transcriptomics to define novel promising targets for SARS-CoV-2.

**Keywords:** aptamers; artificial intelligence; SARS-CoV-2; spike receptor; AXL; therapeutics; single-cell transcriptomics





## 1. Introduction

The ongoing coronavirus disease 2019 (COVID-19) pandemic caused by the new SARS-CoV-2 virus was initially identified in December 2019 in Wuhan, China [1,2]. Worldwide, the number of COVID-19 fatalities has been relatively high among the elderly and patients with cardiovascular disease and/or arterial hypertension.

Respiratory, intestinal, hepatocellular, and neurological disorders are caused by CoVs, which are RNA viruses that infect humans and other animals, such as mammals and birds. Six CoV strains have been identified as being able to cause human diseases. SARS-CoV (2002 and 2003; China), Middle East respiratory syndrome-related CoV (2012; the Middle East), and SARS-CoV-2 cause fatal illnesses characterized by systemic inflammatory reactions [1].

Worldwide, as of March 2023, close to 761 million cases and >6.87 million deaths were reported, including >100.3 million cases and >1113 million deaths in the US and 274 million cases and >2.93 million death in Europe related to SARS-CoV-2.

As with the other coronaviruses (CoVs), the entry of SARS-CoV-2 into host cells is mediated by the transmembrane (TM) glycoprotein spike (S), which is expressed in the SARS-CoV-2 envelope. The SARS-CoV-2 S protein binds to angiotensin-converting enzyme 2 (ACE2), which is widely expressed in a large variety of host cells, inducing virus internalization [3]. Recently, several studies have demonstrated that other receptors are involved in virus entry into host cells. Among them, the AXL, a receptor tyrosine kinase, facilitates virus entry as powerfully as ACE2 [4,5]. Silencing AXL significantly reduces pulmonary infection by SARS-CoV-2 [5]. Targeting AXL, therefore, may be a therapeutic strategy for SARS-CoV-2.

One of the groups with the highest risk of SARS-CoV-2 infection is cancer patients (*n* = 18 million globally according to the World Health Organization in 2020), owing to their intrinsically weakened immune systems as well as their cancer treatments. Researchers found that they were 25–55% more susceptible to COVID-19 than cancers depending on the cancer type, with hematological and lung cancer patients having the highest susceptibility [6]. Additionally, leukemia and lymphoma patients had the highest COVID-19 mortality rate at 37%; for patients with solid cancers, the mortality rate was 25%. Investigators observed differences in mortality rates among patients with specific solid cancers. For example, the mortality rates were 55% for patients with lung cancer, 38% for those with colorectal cancer, 14% for those with breast cancer, and 20% for those with prostate cancer [7].

This poses a specific and additional immediate burden on cancer patients who are already facing the tribulations of treatment and survivorship and would force changes in current treatment strategies. SARS-CoV-2 will continue to be a major threat to the lives of the above high-risk groups, including current and future cancer patients, irrespective of the type and stage of cancer unless treatment is developed urgently.

Early detection and treatment of SARS-CoV-2 infection represent the front line in controlling its spread. Few drugs are approved for the treatment of COVID-19, so the development of novel therapeutic strategies is needed [8].

For meaningful medical applications of tools for treatment, reliable diagnosis, and effective therapy are essential. Hence, attempts to produce small compounds with the ability to control target activities have increased. Recently, aptamers were developed to achieve desired outcomes in connection with medical applications. Aptamers are promising therapeutic tools for the development of novel therapeutic and diagnostic methodologies. They can bind to and regulate the activity of proteins based on their tertiary structural interactions [9]. Aptamers can target a wide range of molecules, including small ones (amino acids, nucleotides, and antibiotics) [10] and large ones (e.g., proteins [11], viruses, and bacteria [12]), as well as whole cells [13]. Various DNA and RNA aptamers that target whole viruses, extracellular viral proteins, or capsid proteins have been isolated. These aptamers function as biosensors or antagonists to inhibit viral replication [14]. The secondary and tertiary structures of aptamers are fundamental to their ability to recognize and tightly bind to target molecules. Aptamers can be considered "synthetic antibodies" with comparable or better binding activity than antibodies and with fewer or no immunogenic effects [15]. In addition to their lack of off-target immune effects, aptamers are easy to manufacture and are amenable to chemical modifications and/or conjugations with other therapeutics' moieties and can be delivered with high specificity and effectiveness [9]. Restrictions regarding the use of aptamers as therapeutic tools include cross-reactivity, limited availability, fast breakdown by nucleases in biological fluids and tissues, and elimination via renal filtration. Our group designed and chemically modified aptamers (e.g., thio- and fluoro-modifications of nucleotides) and increased their stability, specificity, bioavailability, and functional reactivity [16,17].

In this review, we describe the current state of the selection of aptamers using artificial intelligence (AI) as a promising tool to treat SARS-CoV-2 infections. We also discuss the ability of aptamers to recognize S protein and novel potential targets, such as the AXL receptor and other receptors, that promote SARS-CoV-2 virus entry into host cells. We will summarize the latest reports on single-cell transcriptomics on SARS-CoV-2 to define potential novel targets for the development of novel aptamers.

## 2. Aptamers

Aptamers are synthetic single-stranded DNA or RNA oligonucleotides with nucleotide numbers ranging from 10 to 100. Like antibodies, aptamers can detect target molecules at several concentrations ranging from micromolar to picomolar [18]. In 1990, Tuerk and Gold [19] created an aptamer to attach the T4 DNA polymerase. The same year, Ellington and Szostak [10] coined the term aptamer to describe an RNA molecule having

the capacity to bind to organic colors. An aptamer that attaches to natural dyes was extracted from more than 1000 distinct pool variants. As a result of their propensity to have certain structures, aptamers typically readily generate complementary base pair sets. They can fold into various secondary structures, including G-quadruplexes, internal loops, stems, pseudoknots, bugles, kissing complexes, and tetra loops. These secondary structures may then be combined to generate distinctive complex 3D structures, which can molecularly recognize their corresponding targets in a precise manner [20]. The binding affinity and selectivity of aptamers are produced by combinations of interactions, including base packing of aromatic rings, ionic bonds, van der Waals force, symmetry in the geometric form of molecular, and noncovalent interactions [21].

Many aptamers have been used as effective tools for the diagnosis and therapy of several diseases, including infectious and cancer, since their identification [22]. Although most therapeutic aptamers' targets are extracellular, researchers were able to employ aptamers against cell surface molecules as targeting moieties for drug administration thanks to the discovery that these molecules may be selectively internalized in target cells [23]. The creation of therapeutic drugs that can solely target damaged cells while minimizing toxicity and adverse effects on normal tissues in patients with compromised immune systems continues to be a difficult problem. To do this, several methods for conjugating therapeutic compounds with ligands that facilitate the active binding of the aptamer to target cells have been developed [24]. A case in point is the recent development of aptamers as drug carriers, which have several benefits over other targeting reagents, such as monoclonal antibodies and proteins. Several aptamer-based conjugates for the targeted delivery of secondary compounds have been developed thanks to their high level of specificity and affinity for binding capacity, readily variable chemical nature, and potential to be created against targets with different natures. Many compounds, such as chemotherapeutic drugs, small interfering RNAs (siRNAs), nanoparticles, microRNAs (miRNAs), peptides, and even other aptamers, have been combined with functionalized aptamers. These hybrid compounds, also known as aptamer chimeras, can precisely attach to target cells and transport their contents [25].

Aptamers are designed and selected through an iterative process called systematic evolution of ligands by exponential enrichment (SELEX) [10,18]. Briefly, SELEX consists of cyclic steps of iterative binding of the random starting pool of oligonucleotides (containing $10^{14}$–$10^{15}$ variants of random sequences, with a central random region flanked by two constant regions) to purified target molecules followed by separation and amplification of the oligonucleotide sequences with the highest affinity for the protein target.

Although SELEX is a well-established technology, novel and effective strategies for identifying aptamers against specific targets have been developed, increasing the efficiency of aptamer selection. One of the most recent emerging technologies for Aptamer–Target Binding Prediction is the identification of aptamers with artificial intelligence (AI). AI can quickly select and identify aptamers against specific targets from a plethora of sequences among millions of different sequences based on aptamer characteristics that were predicted from aptamer–target structure analysis. In the section below we summarized the use of AI for Aptamer–Target Binding Prediction and the selection and use of aptamers as diagnostics and therapeutics for viral infections.

## 2.1. Artificial Intelligence: An Emerging Approach to Aptamer Selection

Aptamer selection is based on the in vitro SELEX approach, which was first performed in 1990 by two independent groups [10,19]. SELEX is based on cyclic rounds of incubation of random nucleic acid libraries to target molecules. The target–nucleic acid complex is isolated from the unbound nucleic acid library pool, amplified, and sequenced. The entire process takes 8 to 20 rounds. However, many pitfalls are encountered during the SELEX process. Therefore, more efficient and easier methods of selecting aptamers are needed. Among these methods, the latest to be discovered is AI, representing a promising approach

to expanding insight into pools using multi-omics data sets [26]. AI uses machine/deep-learning algorithms to predict potential targets.

There are difficulties associated with using SELEX technology. Aptamers are often developed using SELEX, which was originally used in the 1990s [10,19]. The use of SELEX technology comes with some challenges. For example, obtaining potential aptamer candidates can take weeks or even months, the efficiency scores for aptamer candidates usually are very low, and only a small number of representative aptamer candidates from the next-generation sequence (NGS) information can be synthesized for binding and affinity characterization. Recently, owing to their simplicity and affordability, several researchers have chosen computational approaches for the identification of aptamer candidates [27,28]. These techniques are used to estimate the aptamer affinity for potential targets using the protein structural data [29]. Several computational tools are available online and have been used for estimating the secondary structure and 3D architecture of RNAs and DNAs, including RNAfold and RNAComposer [30]. These tools have also the potential to identify the structural motif of aptamers similar to the analysis of other RNAs and DNAs. Molecular docking and molecular modeling methods, which are often used to select protein compounds based on structural information for selection, have demonstrated efficiency in the selection of protein–aptamer targets [31].

New computational approaches to selecting aptamer candidates with high affinity and specificity for target molecules in drug development have been inspired by AI, particularly machine/deep-learning algorithms [32]. Several machine/deep-learning techniques have been found to perform better than a variety of conventional approaches in predicting binding affinity, including molecular docking and computational techniques [33]. For example, researchers used AI and a random forest model to create tiny compounds that selectively target microRNA–mRNA interactions [34]. Machine/deep-learning approaches show promise in aptamer–target affinity prediction, although they are not widely employed in aptamer-based discovery at the moment. Machine/deep-learning algorithms can be effectively used to examine far greater volumes of experimental data than the conventional model because they do not require the structural knowledge that the use of aptamers requires. Furthermore, the use of larger study data sets may enhance the effectiveness of these machine/deep-learning approaches [33].

### 2.2. Aptamers against the Virus: Therapeutic and Diagnostic Applications

In the design of sensing devices to identify various pathogens, aptamers can be used as targeted agents for the administration of drugs [35]. Additionally, aptamers can bind to viral particles implicated to have roles in different viral infection phases [36]. Particularly, several aptamers have been created to identify crucial intrinsic or metabolic characteristics related to bacterial and viral diseases [37]. Aptamers are beneficial as biosensors because of their selectivity toward therapeutically important indicators. Aptamers for a wide range of viruses have been generated in recent years, including the vaccinia virus, dengue virus (DENV), hepatitis C virus (HCV), human immunodeficiency virus (HIV), apple stem pitting virus, norovirus, rabies virus (RABV), bovine viral diarrhea virus, hepatitis B virus (HBV), Ebola virus, and influenza virus [38,39]. Aptamers are integrated with a number of viral detection technologies, including fluorescent, radioactive, electrical, chemical, photonic, spectrophotometric, and enzyme-linked analyses, in order to use molecular interactions for the monitoring of aptamers. Aside from their ability to identify, aptamers can prohibit proteins or other molecules from attaching to their intended targets and thus suppress the spread of viral infections by inhibiting viral reproduction [40]. Some of the virus-targeted aptamers that have been used in detection are listed in Table 1.

Recently, Yamamoto et al. [41] created an RNA aptamer that specifically targets the HIV-1 trans-activator of transcription (Tat) protein. This protein, one of the HIV auxiliary molecules, is crucial for the transcription of viral RNA and improves the quantity of protein generated by binding to the viral RNA. Because the Tat protein is secreted by an organism at the onset of infection, it is one of the most promising choices for HIV

screening. In subsequent research, Minunni and colleagues produced aptamer-based sensors for tracking Tat protein [42]. Another study demonstrated that several aptamers against the gp120 component of the HIV-1 coat glycoprotein are accessible. One of these gp120 RNA aptamers, B40, prevents gp120 from attaching to its T-lymphocyte co-receptor C-C chemokine receptor type 5 [43]. In 1994, Wyatt and collaborators demonstrated that a phosphorothioate guanosine-quartets oligonucleotide ($T_2G_4T_2$) strongly inhibited HIV entry by blocking the virus–cell membrane fusion initiated by the V3 region of the gp120 protein, which contained the high percentage of positively charged amino acids across all the HIV variants [44]. In 1990, Hotoda and coworkers discovered the active sequence d(TGGGAG) was able to inhibit the HIV virus in the micromolar range. This 6-mer sequence d(TGGGAG) was defined as an aptamer able to recognize the V3 and CD4-binding site of gp120 of the HIV virus. In the same year, Hotoda and collaborators built a library based on sequence d(TGGGAG) conjugated with different aromatic residues. They found that the more potent inhibitor was R-95288, a 5′-end modified 6-mer with $IC_{50}$ of 1.0 μM and low toxicity [45–47]. Further studies on the 5′-end modified 6-mer d(TGGGAG) clarified the structure relationship of the G-quadruplex oligonucleotides and the greatest anti-virus activity [48–50].

Cirrhosis of the liver, hepatocellular cancer, and chronic hepatitis are all often driven by HCV. Aptamer research aimed at the detection and treatment of HCV infection remains in its infancy, and an authorized HCV vaccine is lacking. Chen et al. [51] used cell surface SELEX to create a single-stranded DNA aptamer called ZE2 that binds selectively to the HCV surface glycoprotein E2 to prevent HCV from binding to host cells and to provide tools for the early identification of HCV infection. In another study, Romero-López and colleagues created multivalent aptamers that target the HCV gene. They suggested that the HCV IRES domains are concurrently targeted by the chimeric RNA aptamer known as HH-11 [52].

In mammalian and bird species, influenza viruses cause respiratory infections. Of influenza, A, B, and C virus subtypes, the influenza A virus is considered to be the primary cause of acute infectious illness and fatalities because it can generate seasonal epidemics as well as rare pandemics [53]. The primary target for aptamers in the cell is the hemagglutinin (HA) antigen, which is widely expressed on the surface of influenza viruses. Moreover, the binding of an influenza virus to a species-specific host cell in the respiratory epithelium is facilitated by the HA antigen. Almost as many aptamers against HA have been created thus far for diagnostic as therapeutic uses [54–56].

Even though aptamers have been used to treat several viral infections for more than two decades, research on their use for this purpose is still in its infancy and has not been managed effectively. Human papillomavirus, HBV, DENV, and RABV are all subjects of current investigations of aptamers. Many aptamers specifically bind to viral surface antigens or virus-infected cells. For example, the RNA aptamer S9 specifically binds to HBV truncated P protein and inhibits binding to this P protein [57]. Additionally, the single-stranded DNA aptamer S15 targets the DENV-2 envelope domain protein [58], and the single-stranded DNA aptamer GE54 targets neurotropic RABV glycoprotein-expressing cells and RABV-infected cells [59,60].

**Table 1.** Aptamers used against viruses.

| Aptamer Type | Aptamer Name | Molecular Target | Organism | Effect | Reference |
|---|---|---|---|---|---|
| Phosphorothioate guanosine-quartets oligonucleotide | $T_2G_4T_2$ | V3 region of the gp120 protein | HIV | Blocking the virus–cell membrane fusion | [44] |
| 6-mer sequence d(TGGGAG) | R-95288 | V3 and CD4-binding site of gp120 | HIV | Anti-virus activity in the range of micromolar and low toxicity | [45–50] |
| RNA | RNA-Tat | TAT | HIV | Even in the presence of a significant abundance of HIV TAR in cell culture, exhibits strong binding to Tat | [41] |
| RNA | B40 | Gp120 | HIV | Blockage of the interaction between gp120 and C-C chemokine receptor type 5 | [43] |
| DNA | T30695 | Integrase | HIV | HIV-1 integrase inhibition | [61] |
| 2′-DEOXY-2′-FLUOROARABINONUCLEOTIDE (FANA) | FA1 | RT | | HIV RT inhibition | [62] |
| RNA | G6-16 | N53 protein | HCV | N53 inhibition | [63] |
| DNA | ZE2 | E2 | HCV | Inhibition of HCV in vitro | [51] |
| DNA | C4 | Core protein | HCV | Inhibition of HCV | [64] |
| DNA | E10 | HA | Influenza A virus (H5N1) | Receptor-binding inhibition | [54] |
| DNA | HA12-16 | Glycosylated HA | Influenza A virus (H5N2) | Prevention of influenza virus infection | [55] |
| RNA | S9 | Truncated P protein | HBV | Inhibition of P protein binding | [65] |
| DNA | S15 | Envelope protein domain III | DENV-2 | Inhibition of proliferation | [58] |
| DNA | GE54 | Glycoprotein | RABV | Inhibition of viral replication | [59] |
| RNA | G5 α3N.4 | E7 protein | Human papillomavirus 16 | E7 Inhibition | [66] |

### 3. Severe Acute Respiratory Syndrome Coronavirus SARS-CoV-2

The CoVs are enveloped with a positive-sense single-strand RNA genome and belong to the *Coronaviridae* family. CoVs are composed of at least four major fundamental proteins: S, envelope protein, membrane protein, and nucleocapsid (N) protein [67]. S protein mediates host binding and subsequent virus–cell membrane fusion during viral infection [68]. Consequently, the S protein determines the diversity of the cell tropism or host range that the virus can infect. Therefore, the role of the S protein offers a great opportunity to target SARS-CoV-2 therapeutically to interfere with virus entry and stop it from infecting the host cells.

N protein, a major structural protein, binds directly to the CoV RNA genome, thus providing stability around the enfolded nucleic acid and creating a shell or capsid. N protein also binds to the viral membrane protein during viral assembly and assists in RNA synthesis and folding. Moreover, it plays a role in virus budding and affects host cell responses, including cell cycle progression and translation [69].

Coronaviruses enter host cells via binding of the virus surface unit, S1, of the S protein to the cellular receptor, ACE2, which facilitates viral attachment to the surface of target cells [70]. ACE2 is involved in the control of systemic arterial pressure [71]. The binding of SARS-CoV-2 using its S protein leads to severe respiratory and cardiac problems. Thus, COVID-19 drugs must avoid any issues with ACE2. S protein priming is performed by the cellular serine protease TM protease serine 2 via cleavage at S1/S2, which allows for the S2 subunit to drive the fusion of viral and cellular membranes [72], leading to respiratory, enteric, hepatic, and neurological damage. Based on the crystal structure of CR3022, a neutralizing antibody previously isolated from a convalescent SARS patient, in a complex with the SARS-CoV-2 S protein receptor-binding domain (S-RBD), the S-RBD spans amino acids 333 and 527 of the S protein [73]. The goal is to target the S-RBD of S protein and stop the entry of SARS-CoV-2 into host cells and thus cause an infection.

#### 3.1. Spike Receptor: Its Structure and Mechanism of Binding

SARS-CoV-2 is an RNA virus with a single-stranded envelope [74]. Its entire genome, measuring 29,881 bp in length and containing 9860 amino acids (GenBank no. MN908947), was identified using an RNA-based metagenomic next-generation sequencing method [75]. Both structural and nonstructural SARS-CoV-2 proteins are generated by gene fragments. S, envelope, membrane, and N genes provide instructions for making structural proteins, whereas the open reading frame area produces nonstructural proteins such as 3-chymotrypsin-like protease, RNA-dependent RNA polymerase, and papain-like protease [76].

The surface of SARS-CoV-2 is covered by a high number of glycosylated S proteins that attach to the ACE2 receptor on host cells to assist viral cell entrance (Figure 1) [77]. The S protein, which is 180–200 kDa in size, consists of a TM segment anchored in the viral membrane, an extracellular N-terminus, and a short cellular C-terminal section [78]. The SARS-CoV-2 S protein has a total of 1273 amino acids and is composed of the S1 and S2 subunits, which are in charge of binding to receptors and fusing membranes, respectively. The N-terminal domain and receptor-binding domain of the S protein are both in the S1 subunit. The S2 subunit is composed of the fusion peptide, heptapeptide repeat sequences 1 and 2, TM domain, and cytoplasm domain [79]. The trimer of the S protein, which forms a bulbous crown on the virus, envelops the virus. The S1 and S2 subunits constitute the bulbous head and stalk part of the S protein [80].

A key component of SARS-CoV-2 infection is the S protein. TM protease serine 2, which is found on the host cell membrane, activates the S protein when the protein attaches to the ACE2 receptor, facilitating virus entrance into the cell [81]. The S protein of SARS-CoV-2 mediates receptor interaction, cell adhesion, and fusion during infection similar to other CoVs (Figure 1) [82–84].

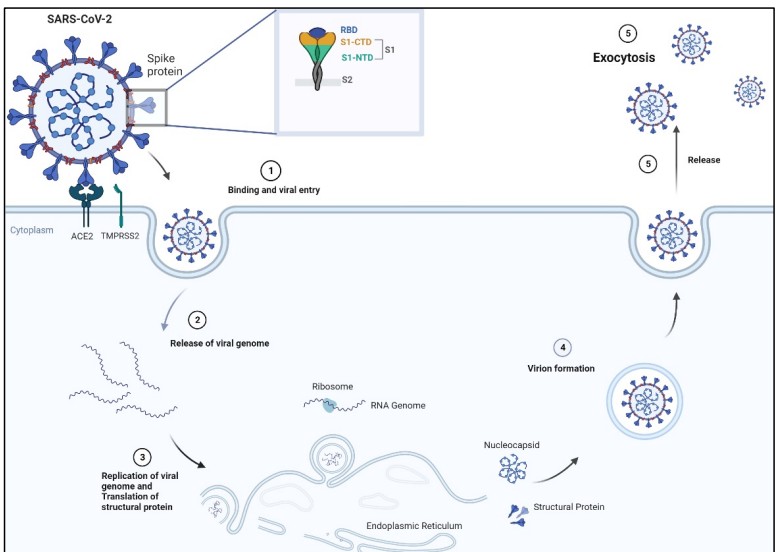

**Figure 1.** ACE2 mediated the life cycle of SARS-CoV-2 to enter inside the host cells. Coronavirus structure showing spike (S) structure. The receptor-binding domain (RBD); S1 subunit composed of S1-CTD: C-terminal domain and S1-NTD; N-terminal domain and S2 subunit. Virus binding to the ACE2 receptor (1); the viral genome is released into the host cell cytoplasm (2); starting to replicate the viral genome in the ribosome and translate the structural protein in the endoplasmic reticulum (3); virion formation occurs (4); and release via exocytosis (5).

### 3.2. Aptamers against Spike Receptors

A viral infection is caused by the binding of viral proteins to the surface receptors on the host cell. Thus, receptor identification is a crucial factor in determining viral entrance and a target for therapeutic development [85]. Unlike the other structural proteins, the S protein exhibits many characteristics that make it a good target for SARS-CoV-2 infection diagnosis, vaccines, and treatments [82,86,87].

The S protein is the most widely used protein for aptamer synthesis because of three characteristics. First, because a significant portion of this protein is present on the surface of the virus, the destruction of the virus is not necessary for an aptamer to recognize it or for the design of diagnostic procedures using an aptamer. Second, because the S protein is a trimeric protein consisting of three identical monomers, it can be engineered to bind to two or three components of the same S receptor with high affinity. Third, the S protein is present in several copies on the surface of each viral particle, offering yet another method of multivalent recognition and high binding efficiency [88]. In recent years, several S protein-targeted aptamers have been designed by many researchers, which are listed in Table 2.

Recently, Song and colleagues identified and selected aptamers that recognize or bind to the RBD of the S protein [89]. They discovered two DNA aptamers with a variety of binding affinities using AI and machine-learning approaches. The researchers used flow cytometry, modeling, and a His-tag modified Ni-bead model/system to track the interactions between the aptamers and the SARS-CoV-2 S-RBD. Both aptamers partially obstructed and occupied the SARS-CoV-2 RBD's ACE2 interaction region. The use of aptamer multimerization techniques and logical mixes or cocktails of aptamers would significantly increase their binding effectiveness and boost their SARS-CoV-2 neutralizing ability to block possible viral escape mutation development. For instance, Sun et al. [90] showed that aptamer cocktails outperformed their single aptamer equivalents at the same concentration in terms of deactivating RBD and pseudovirus. Furthermore, Valero et al. [91] used a framework or poly(A)-linking strategy to create dimeric and trimeric forms of RBD-PB6-Ta aptamers.

Both the dimeric and trimeric variants of an aptamer exhibited better S protein binding avidity than the monomeric form, and they both effectively neutralized the genuine SARS-CoV-2 virus with corresponding half-maximal inhibitory concentrations.

Additionally, the SARS-CoV-2 virus was inhibited and destroyed by six extremely effective novels DNA aptamers, according to Huang and colleagues. These aptamers prevented RBD-ACE2 interaction while exhibiting significant S protein S1 specificity and affinity. Of note is that the lead aptamer recognized the S1 protein in serum and decreased the incidence of suppressed SARS-CoV-2 infections in a pseudoviral system.

Song et al. [89] were the first to synthesize aptamer targeting the RBD-binding site of the S protein using the bead-based SELEX method. Numerous studies have identified RBD-binding DNA and RNA aptamers that can effectively prevent RBD binding to ACE2 using identical SELEX techniques [90–92].

Yang et al. [93] showed that the ST-6, ST-6-1, and ST-6-2 aptamers have a high affinity for the S1 and trimeric S protein of SARS-CoV-2. In addition, Zhang and colleagues designed an MSA52 aptamer using the electrophoretic mobility shift assay-based SELEX method, showing that this aptamer has a high affinity for the S1 protein [94].

**Table 2.** Aptamers against the SARS-CoV-2 virus.

| Aptamer Type | Aptamer Name | Molecular Target | SELEX Method | Reference |
|---|---|---|---|---|
| DNA | CoV2-RBD-1C CoV2-RBD-4C | RBD | Bead-based | [89] |
| DNA | CoV2-6C3 cb-CoV2-6C3 | RBD | Bead-based | [90] |
| DNA | Aptamer-1 Aptamer-2 | RBD | Bead-based | [92] |
| DNA | nCoV-S1-Apt1 | RBD, S1 protein | Capillary electrophoresis-based | [8] |
| DNA | MSA1 MSA5 | RBD, S1 protein, trimeric S protein | EMSA (electrophoretic mobility shift assay and magnetic bead-based) | [95] |
| RNA | RBD-PB6 | RBD | Bead-based | [91] |
| DNA | XN-268s | S1 protein | Magnetic bead-based | [96] |
| DNA | ST-6 ST-6-1 ST-6-2 | Trimeric S protein | Bead-based | [93] |
| DNA | S2A2C1 S1B6C3 | S2 protein, RBD | Bead-based | [97] |
| DNA | MSA52 | S1 protein | Electrophoretic mobility shift assay, EMSA-based | [94] |

### 3.3. Aptamers as Diagnostics for SARS-CoV-2

The COVID-19 pandemic forced scientists to develop rapid diagnostic methods to limit the spread of infection [98]. Currently, several tests have been developed that detect SARS-CoV-2, including RT-PCR-based tests, tests quantifying IgM and IgG antibodies directed against the virus, and tests detecting virus antigens. Each of these methods has several advantages and disadvantages (Table 3). The RT-PCR method is based on the primer probe sets targeting RdRp, Orf1, or the N region of SARS-CoV-2 [99,100]. Although RT-PCR tests have the highest specificity, laboratories are only able to perform a certain number of tests per day. In addition, in this method, patients who have recently contracted SARS-CoV-2 may obtain a false-positive result because the virus has not been completely removed from the body, and patients in the early stages of the disease may obtain a false-negative result. Serological tests provide only information about the production of antibodies by the body, but unfortunately, they are not direct diagnostic tests. These tests are based on LFA and ELISA technology [100,101]. Among all COVID-19 tests, antigen tests have the lowest sensitivity and specificity. However, they are one of the fastest and simplest methods [100,101].

**Table 3.** A comparison of diagnostic methods of the SARS-CoV-2 virus.

| Assay | Mechanism | Specificity and Sensitivity | Advantages | Disadvantages | References |
|---|---|---|---|---|---|
| RT-PCR | Based on the polymerase chain reaction | Sensitivity: 92.7% (4–5 days after infection), 88% (5–14 days after infection), and specificity ~100% | The best available diagnostic test enables the qualitative and quantitative determination of the COVID-19 virus in the downloaded material | -Expensive<br>-Risk of false negative or false positive results<br>-Requires special equipment<br>-Not fast as antigen tests | [102,103] |
| Antibody | Monoclonal antibodies detect the host's antibodies (usually directed against two virus proteins, S and N) | Depends on the detected antibody; average specificity: 75–90% and sensitivity: 77–90% | Helps to detect who previously had COVID-19 or evaluate how the vaccine affected the immune response | -Gives information only about the number of antibodies in the patient's blood<br>-Does not provide information about how strong the infection is<br>-Does not detect antibodies in early stages of the disease<br>-The level of antibodies does not correspond to the stage of the disease<br>-Risk of cross-reactions and similarity to the SARS-CoV-2 virus sequence to other viruses | [101,104,105] |
| Antigen | Detect SARS-CoV-2 proteins | Specificity: 55–100%<br>Sensitivity: 30–100% | Fast time to obtain results, does not require expensive equipment, and is cheap | -False-positive results in patients without symptoms<br>-Accuracy of tests is unknown<br>-These tests are not approved by the WHO for diagnosis | [106] |
| Aptamers | Recognize S and N proteins of SARS-CoV-2 | Depending on the aptamer, sensitivity, and specificity are comparable with RT-PCR | Cheaper than RT_PCR, detects virus in early stages, detects small amounts of the virus in the sample, does not require extraction and amplification of the virus's genetic material, and low cost of tests | -Expensive cost of synthesis aptamers on a big scale (kilograms) | [107–109] |

Despite the high number of COVID-19 tests available on the market, there is still no effective and unequivocal diagnostic method for the SARS-CoV-2 virus. This is prompting scientists to look for new detection technologies, including aptamers [110]. Currently, we can distinguish various types of tests utilizing aptamers including enzyme-linked aptamer binding assays (ELAAs), aptamer-based lateral flow assays (LFAs), and aptasensors. These methods can detect S protein, N, and virus particles [101,110]. Sharma et al. presented a method of SARS-CoV-2 detection using aptamers with high specificity (99%) and sensitivity (91%). The detection parameters were comparable with RT-PCR methods. The designated aptamer is targeting spike trimeric protein. Moreover, this diagnostic test is a rapid test because results can be obtained in three hours [111]. In 2023, Li et al. designed a biosensor based on the aptamer to detect COVID-19 S protein. The device consists of thiolated aptamers recognizing the S protein of SARS_VoV-2 incorporated on the gold nanoparticle film—SCAP (spherical cocktail aptamers–gold nanoparticles)—and SERS (Surface-enhanced Raman Spectroscopy) nanoprobes. SERS is a combination of gold nanoparticles, Raman reporter molecules, and aptamers recognizing S protein. The demonstrated platform shows high specificity against the SARS-CoV-2 virus with a sensitivity of 97.5% and a specificity of around 100% [112]. Furthermore, in this test, the cost of testing one sample with a demonstrated device is 32 times lower compared to RT-PCR [112,113]. This shows how much potential aptamers have in the diagnosis of the SARS-CoV-2 virus and more.

## 4. Novel Receptors Promote SARS-CoV-2 Entry into Host Cells

According to scientific opinion, ACE2, which is mainly expressed in the human kidney and digestive tract, is the principal receptor that SARS-CoV-2 uses to enter human cells. However, ACE2 expression levels are low in humans, particularly in the respiratory tract, such as in lung and tracheal cells. This does not support the high infectiousness of SARS-CoV-2. Therefore, other receptor molecules may be involved in the entry of SARS-CoV-2 into the human respiratory system (Figure 2) [114]. Recent studies demonstrated that SARS-CoV-2 binds to host receptors that are widely expressed on the surface of host cells [115].

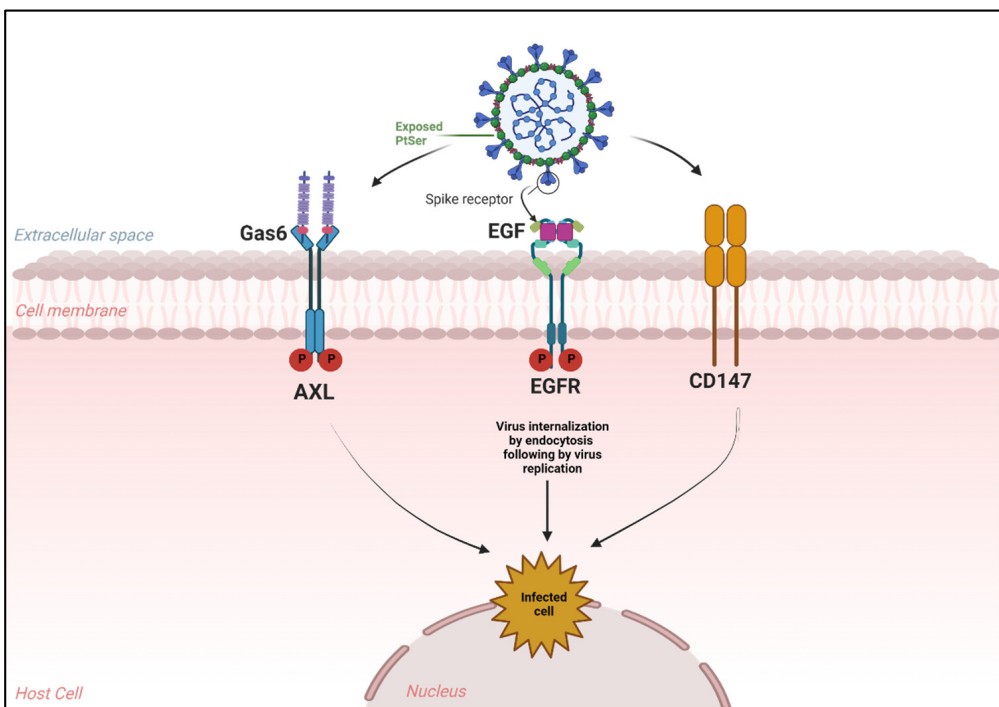

**Figure 2.** Schema of the interaction between alternative receptors that mediate SARS-CoV-2 entry.

### 4.1. Alternative Receptors That Mediate SARS-CoV-2 Entry

AXL is a member of the TAM receptor tyrosine kinase family, which also includes the TYRO3 and MER receptors. The AXL receptor has three domains: the extracellular domain, which is the binding site for the AXL receptor ligand; the TM domain in a single-chain alpha helix structure; and the intracellular domain with tyrosine kinase activity [116]. Gas6, a γ-carboxylate protein, binds to AXL with high affinity and is the main ligand for AXL [117]. AXL is expressed in many normal cells and tissues, such as macrophages, endothelial cells, the heart, the liver, and skeletal muscle. However, it is overexpressed and activated in many human cancers, such as breast, lung, ovarian, and pancreatic [118]. In addition, AXL, which is known to have biological functions such as cell growth, cell migration, aggregation, metastasis, and cell adhesion, has an important role in the development of infectious diseases.

Recently, Wang and colleagues discovered that the N-terminal region of the SARS-CoV-2 S protein interacts with AXL [4]. They showed that in HEK293T cells, the overexpression of AXL enhanced SARS-CoV-2 entry as effectively as the overexpression of ACE2, whereas knockout of AXL considerably decreased SARS-CoV-2 entry in H1299 pulmonary cells and human primary lung epithelial cells. They also found that the three major potential receptors with the best affinity ratings were AXL, EGFR, and low-density lipoprotein receptors because of their protocol using next-generation protein–protein interaction methods to identify candidate receptors for viral entry.

Another study by Bohan et al. [119] demonstrated that when low ACE2 concentrations are present in cells, AXL and other phosphatidylserine receptors enhance SARS-CoV-2 virus infection. They also observed that the AXL inhibitor bemcentinib inhibited SARS-CoV-2 infection in a variety of lung cell lines, including some with low ACE2 expression, but did not have an inhibitory effect on infection in Calu-3 cells.

Boytz et al. [120] discovered that SARS-CoV-2 infection efficiency was reduced in an A549-ACE2 cell line in which they used CRISPR-Cas9 technology to knock out AXL. Overall, as seen in other previous studies, AXL expression levels were associated with SARS-CoV-2 infection. Therefore, AXL may be a target receptor protein in therapy for SARS-CoV-2 infection.

Although ACE2 expression is critically important to SARS-CoV-2 infections, in some cases, its expression is not related to the infection or the clinical profile. In addition, SARS-CoV-2 infection of human cells lacking ACE2 demonstrates that these cells have alternative receptors that can mediate virus entry. Most of these receptors, including the cluster of differentiation 147 (CD147), asialoglycoprotein receptor 1 (ASGR1), low-density lipoprotein receptor class A domain-containing 3 (LDLRAD3), and Kringle containing TM protein 1 (KREMEN1), are known to be involved in the entry of other viruses and affect cellular functions and immune responses. Additionally, EGFR is a TM protein and a member of the receptor tyrosine kinase family. These receptors regulate many cellular processes by acting on signaling pathways, such as phosphoinositide 3-kinase/AKT and Ras/Raf, through phosphorylation.

EGFR upregulation increases cell migration and proliferation in many cancer types. Overactivation of EGFR during viral infections has an important role in the entrance of the virus into cells and triggers the inflammatory response. Researchers showed that the development of micropinocytosis during SARS-CoV-2 infection is dependent on the activation of EGFR. In another study using a mouse model of SARS-CoV pathogenesis, investigators showed that EGFR has a critically important role in lung tissue damage and fibrosis caused by SARS-CoV. In more recent studies, CD147, also known as basigin, played a role as a receptor for SARS-CoV-2 entrance into cells with low ACE expression. However, reports of CD147's function as a receptor for SARS-CoV-2 entrance are conflicting.

Wang et al. [121] demonstrated that CD147 mediates viral entry by interacting with the SARS-CoV-2 S-RBD in cells with low ACE2 expression. Another study demonstrated that CD147 silencing reduced ACE2 expression and SARS-CoV-2 viral N1 RNA levels in host cells. Shilts et al. discovered that CD147 knockdown did not influence the susceptibility

to SARS-CoV-2 infection in Calu-3 cells with high ACE2 expression. In light of all of this information, CD147 is thought to act as an alternative SARS-CoV-2 entry receptor in cells with low ACE2 expression, but the receptor function does not have a pronounced effect on cells with high ACE2 expression [122]. Studies have demonstrated that the C-type lectin types of the cluster of differentiation 209 ligands (CD209L), liver/lymph node-specific intercellular adhesion molecule-3 grabbing integrin, the cluster of differentiation (L-SIGN), dendritic cell-specific intercellular adhesion molecule-3 grabbing nonintegrin (DC-SIGN), and C-type lectin domain family 4 member G are alternate receptors for SARS-CoV-2 [123].

In addition, genome-wide screening of CRISPR activation revealed low-density lipoprotein receptor class A domain-containing 3 and TM protein 30A to be ACE2-independent SARS-CoV-2 receptors. Several SARS-CoV-2-binding proteins were discovered by Gu et al. using a high-throughput receptor profiling technique. Kringle containing TM protein 1 and asialoglycoprotein receptor 1 were identified as ACE2-independent SARS-CoV-2 receptors after the screening of 5054 human membrane proteins for interactions with the extracellular S and Fc fusion proteins of SARS-CoV-2 [124].

### 4.2. *Aptamers against Those Receptors*

Aptamers have been used to diagnose and treat a variety of infectious diseases, including parasitic, bacterial, and viral infections. Aptamers can bind to a variety of both organic and inorganic particles, including proteins. Aptamers have also been demonstrated to successfully bind to whole viral particles and even cells (Table 4). The development of aptamers against receptors that are effective in the pathogenesis of SARS viruses may be a promising approach to the treatment of aptamers (Figure 3). Amero et al. [16] discovered that AXL aptamers, which are DNA aptamers targeting AXL that they designed and switched from RNA aptamers, reduced tumor development and spread for ovarian cell lines and orthotopic ovarian cancer models. In addition, we synthesized a DNA aptamer-based on the GL21.T sequence of an RNA aptamer. We found that the inhibition of AXL phosphorylation and associated cell proliferation by this AXL aptamer affected the effectiveness of chemotherapy in ovarian cancer models both in vitro and in vivo [17]. Additionally, Cerchia et al. [125] found that the AXL RNA aptamer GL21.T inhibited tumor growth in vivo in a mouse xenograft model created with human non-small cell lung cancer cells.

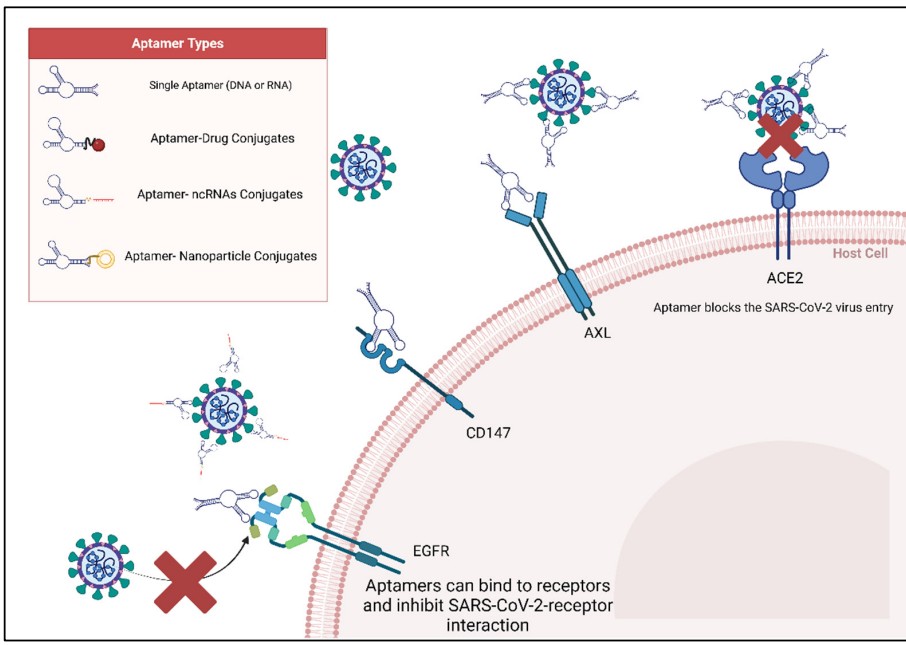

**Figure 3.** Schematic potential for therapeutic approaches using aptamer and aptamer chimera technologies.

Li and colleagues discovered that the anti-EGFR aptamer they designed inhibited the growth of A431 cancer cells [126]. Other studies using anti-EGFR aptamers demonstrated them to be effective against many pathological processes, such as cancer [127,128]. Additionally, aptamer-based methods may be viable approaches for identifying and treating SARS-CoV-2 infections. Aptamers can effectively inhibit the virus's activity in a variety of ways, including as carriers of therapeutic agents. They are nonimmunogenic and bioavailable therapeutic tools. Additionally, aptamer-based treatments can be improved in several ways. Although recognized applications of these particles in the treatment of viral illnesses are lacking, it may be worthwhile in the next and to keep this in mind.

**Table 4.** Aptamers are used against other receptors.

| Aptamer Type | Aptamer Name | Molecular Target | SELEX-Method | References |
|---|---|---|---|---|
| RNA | GL21.T | AXL | Bead-based | [125] |
| DNA | AXL-APTAMER | AXL | Bead-Based | [17] |
| DNA | AXL-APTAMER | AXL | Bead-based | [16] |
| DNA | AXL-APTAMER | AXL | Bead-based | [129] |
| DNA | EGFR APTAMER | EGFR | Bead-based | [126] |
| RNA | CL4 Aptamer | EGFR | Bead-based | [130,131] |

## 5. Single-Cell Transcriptomics Analysis as a Potential Tool to Define Novel Targets of SARS-CoV-2

Clinical manifestations of infection with SARS-CoV-2 can be diverse and depend on many factors including the patient's health status, genetic condition, and age. Therefore, to increase the effectiveness of treatment, it is important to understand the genetic profile of patients. Understanding the genetic profile of individual cells may allow for the development of effective targeted therapy [132]. One of the methods used to analyze the genetic profiles of patients is single-cell transcriptomics [133]. This method allows for the analysis of the level of gene expression in individual cell populations and evaluates the interactions between cells. Moreover, it is a potential tool to discover novel targets for SARS-CoV-2 treatment [134]. Therefore, since the coronavirus pandemic, researchers are using sc-RNA sequencing techniques to understand the mechanisms underlying infection and designing targeted therapy using aptamers.

One of the first single-cell analyses of COVID-19 was performed by Wang et al. in 2020. In this research, peripheral blood mononuclear cells from thirteen COVID-19 patients and five healthy people were analyzed. Patients were divided into three groups based on the severity of the disease: moderate, severe condition, and recovered patients. The researchers based on the gene markers identified in peripheral blood mononuclear cells fourteen types of cells. In the research, they also analyzed the effect of COVID-19 infection on the immunological system. It has been presented that patients with severe conditions had decreased levels of MAIT cells (mucosal-associated invariant T cells) and increased levels of CD14$^+$ monocytes. It is believed that these changes may be causing cytokine storms in COVID-19. Increased levels of C14$^+$ monocytes are caused by pathogenic T cells. Therefore, researchers thoroughly analyzed how the SARS-CoV-2 virus may affect individual T-cell populations. It has been shown that levels of the native state of T cells (CD4$^+$, CD8$^+$, and natural killers) were decreased in patients infected with SARS-CoV-2 while active T cell populations (CD4$^+$ effector, CD8$^+$ effector, natural killer CD56, and natural killer CD160) were elevated compared to healthy donors. Moreover, it was demonstrated that COVID-19 infection led to the exhaustion of T cells including CD4$^+$ effectors, CD8$^+$ effectors, and NK CD160 cells. Researchers also discovered that COVID-19-infected patients had increased response to IFN-$\alpha$, which may be an obstacle to using IFN for the treatment of COVID-19 [135]. The literature data also indicates changes in the B cell population in severe COVID-19 cases. Thus, Cho et al. analyzed the effect of coronavirus on B cell changes using sc-RNA sequencing. Within B cells, researchers distinguished four subpopulations:

plasmablast, plasma cells, memory B cells, and naive B cells. It turned out that during COVID-19 infection, the level of plasma cells and plasmablast increases. The phenomenon is more visible with a worse course of disease [136].

Even though COVID-19 infection may be asymptomatic, in many cases, it may affect the heart, kidneys, lungs, gastric system, and circulatory system. In addition, recovering patients are at risk of a deterioration in their quality of life or even death due to damage to internal organs caused by the disease. Therefore, research conducted around the world using sc-RNA analysis may help to evaluate the impact of the disease on individual organs and their cells, which may help to prevent multi-organ dysfunction caused by the coronavirus [137]. In 2022 Mishra, Atri, and Pathak investigated the correlation between the imbalance of pancreatic secretory proteins caused by COVID-19 and organ dysfunction. In this study, they analyzed pancreas samples of patients with COVID-19 using sc-RNA sequencing. Based on gene expression, nine cell types were distinguished. The researchers discovered that five virus genes were expressed in these cells. In the patient's pancreas samples, scientists observed the overexpression of 102 genes responsible for damage to the pancreas and myeloid leukocyte migration pathways. Moreover, an imbalance of pancreatic secretory proteins was associated with dysfunction of the RAS pathway, which resulted in overexpression of cytokines including TNF-$\alpha$, interleukins -1 and -6, and CAM [138]. Taking into consideration the increased level of pro-inflammatory cytokines, Qu et al. published data from two COVID-19 patients treated with tocilizumab. Tocilizumab is an anti-IL-6 antibody with the potential to be used in the treatment of COVID-19. The sc-RNA analysis revealed that in severe COVID-19 conditions, stage-specific monocyte populations (with higher expression of *ATF3*, *NFIL3*, and *HIVEP3*) may be responsible for causing inflammation during COVID-19. Moreover, it has been demonstrated that monocytes may affect other immune cells, which increases the cytokine storm. However, this result requires further research to confirm it [139]. As has been shown, sc-RNA analysis might be a promising tool to identify new targets for COVID-19 therapy. In 2022, Weilbaecher et al. analyzed bronchoalveolar lavage fluid of SARS-CoV-2 patients and discovered that CD68$^+$ macrophages exhibit increased expression of the *HSPE* gene. Moreover, this phenomenon intensified with the increase in the severity of the disease. The overexpression of *HSPE* leads to increased expression of proinflammatory cytokines including TNF and interleukins -1 and -6. Therefore, in the next step of the research, scientists used roneparstat (an inhibitor of HSPE) in treating COVID-19. The results of the study showed that targeting heparanase might be a promising approach for treating SARS-CoV-2 [140].

This shows how single-cell analysis helps to understand the effect of COVID-19 infection on the single cells in the organism and their interactions. Zou et al. reanalyzed the results of single-cell analysis from Gene Expression Omnibus experiments (GSE145926, GSM3660650, GSE109816, GSE115469, GSE131685, GSE129845). They showed that tissues with high expression of ACE2, TMPRSS2, NRP1, AXL, FURIN, and CTSL have higher potency to be infected by COVID-19. In the lungs, macrophages had higher expression of NRP1, AXL, FURIN, and CTSL. Further studies revealed that an increased number of M1 macrophages corresponds with COVID-19 progress. In macrophages with M1 phenotype, increased activation of KRAS and inflammatory response pathways were observed. Moreover, it has been demonstrated that genes associated with the ribosome pathway might be a potential target for the treatment of COVID-19. Interestingly, the most endangered to SARS-CoV-2 infection in heart tissue were fibroblasts along with smooth muscle cells. Both these cell types had increased expression of ACE2, NRP1, AXL, FURIN, and CTSL receptors. A thorough analysis of the kidney data set showed that duct cells and proximal tubule cells might be potentially infected by COVID-19 because of the high expression of TMPRSS2, FURIN, and NRP1. In liver non-inflammatory cells and LSEC cells, cholangiocytes had a higher risk of being infected by COVID-19 compared to the rest of the liver cell types. Researchers could not identify the most potent cell type in the bladder to be infected. These results show the potential of designing the most effective treatment depending on the expression of receptors in the various cell types [141].

The single-cell analysis provides various information about the interaction between cells and pathways involved in the progression of COVID-19. It may help to design innovative aptamers targeting SARS-CoV-2.

## 6. Conclusions

Since viruses often change and can evade the human immune system, numerous antiviral medications and vaccinations are ineffective. Currently, several drugs are being developed to prevent or treat SARS-CoV-2. Targeting the SARS-CoV-2 proteins' evolutionarily conserved functional and structural parts, which are crucial for the initial phases of viral illness and might be a very successful therapeutic strategy that also lowers the risk of creating escape mutations. The SARS-CoV-2 S glycoprotein is the primary target of specific antibodies during infection and the subject of therapeutic and vaccine design because it is surface-exposed and facilitates entrance into host cells [82]. In addition, it is known that the nucleocapsid protein's C-terminal lysine-rich domain serves as a nuclear localization indication that is crucial for accessing the nucleus. Nucleocapsid proteins are consequently discovered in the cytoplasm and nucleus of SARS-CoV-2-infected cells. By focusing on conserved viral epitopes, universal aptamers with cross-protection against infections by SARS-CoV-2 and its variations might be created.

Currently, in the diagnosis and design of new antiviral therapies, omics research, including genomics, transcriptomics, proteomics, and metabolomics, as well as interactions between cells, is of great importance [134]. These techniques allow for a better understanding of the biological basis of virus infection, including SARS-CoV-2. The sc-RNA sequencing analyses show how the severity of the disease, sex, age, and comorbidities lead to several changes in the body's immune response. In addition, a thorough understanding of the processes occurring under the influence of SARS-CoV-2 may potentially allow the development of therapies to limit the progression of the disease, as well as the development of effective therapeutics.

Exciting new potentials for the identification and treatment of different bacteria have arisen because of recent advancements in aptamer technology. We think that aptamers will be extremely important in the identification and suppression of SARS-CoV-2 when used in conjunction with other molecular diagnostic and therapeutic techniques.

**Author Contributions:** P.A., C.R.-A. and G.L.-B. were involved in the conceptualization of the article. P.A., C.R.-A., S.K.S. and A.S. wrote, reviewed, and edited the article. P.A. supervised. All authors have read and agreed to the published version of the manuscript.

**Funding:** S.K.S. was supported by the Scientific and Technical Research Council of Turkey (TUBITAK) 2219-Research Grant in the frame of the International Postdoctoral Research Scholarship Programme, The University of Texas MD Anderson Cancer Center, 2022–2023.

**Institutional Review Board Statement:** Not applicable.

**Informed Consent Statement:** Not applicable.

**Data Availability Statement:** Not applicable.

**Acknowledgments:** We thank Don Norwood, Department of Scientific Publications, for the critical reading of the manuscript. Figures were created with BioRender.

**Conflicts of Interest:** The authors declare no conflict of interest.

## Abbreviations

| | |
|---|---|
| SARS-CoV-2 | Severe Acute Respiratory Syndrome coronavirus |
| COVID-19 | Coronavirus disease 2019 |
| MERS-CoV | Middle East Respiratory Syndrome coronavirus |
| WHO | World Health Organization |
| S | Spike |

| ACE 2 | Angiotensin-converting enzyme 2 |
| AXL | Anexelekto |
| AI | Artificial Intelligence |
| SELEX | System evolution of ligand by exponential enrichment |
| CoVs | Coronaviruses |
| KREMEN1 | Kringle Containing Transmembrane Protein 1 |
| HCV | Hepatitis C Virus |
| HIV | Human Immunodeficiency Virus |
| TMPRSS2 | Type 2 TM serine protease TM protease serine 2 |
| S-RBD | Receptor-binding domain |
| TM | Transmembrane |
| FS | Fusion Peptide |
| HR1 | Heptapeptide repeat sequence 1 |
| HR2 | Heptapeptide repeat sequence 2 |
| RTK | Receptor Tyrosine Kinase |
| Gas6 | Growth Arrest Protein 6 |
| TAM | TYRO3, AXL, and MERTK Family |
| EGFR | Epidermal Growth Factor Receptor |
| LDLR | Low-Density Lipoprotein Receptor |
| CD147 | Cluster of Differentiation 147 |
| ASGR1 | Asialoglycoprotein receptor 1 |
| LDLRAD3 | Low-Density Lipoprotein Receptor Class A Domain Containing 3 |
| TMEM30A | Transmembrane Protein 30A |
| CD209 | Cluster of Differentiation 209 |
| CD209L | Cluster of Differentiation 209 Ligand |
| L-SIGN | Liver/lymph node-specific intercellular adhesion molecule-3-grabbing integrin |
| DC-SIGN | Dendritic Cell-Specific Intercellular Adhesion molecule-3-Grabbing Non-integrin |
| CCR5 | C–C chemokine receptor type 5 |
| SCAP | Spherical cocktail aptamers–gold nanoparticles |
| SERS | Surface-enhanced Raman Spectroscopy |

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
