# Peer review of "Aptamers as Insights for Targeting SARS-CoV-2"

_biologics, doi:10.3390/biologics3020007_

Round 1

Reviewer 1 Report

There are so many similar reviews about aptamers targeting SARS-CoV-2.  I suggest the authors to rewrite the paper to make it different by highlighting the uniqueness. 

1. For figure 1, the label of RBD is wrong. RBD do not contain NTD. 

2. The section 5, Single-cell Transcriptomics of SARS-CoV-2 , seem unrelated to this paper, which should be deleted. Similar case is some part of section 2.1. 

English is OK

Author Response

Query 1: There are so many similar reviews about aptamers targeting SARS-CoV-2.  I suggest the authors to rewrite the paper to make it different by highlighting the uniqueness.

1) Response:

We thank you the reviewer for the comment. We believe that our review is unique due to the potential use of Artificial Intelligence for aptamer selection, reporting the latest discovery in terms of novel receptors involved in SARS-CoV2 entry in host cells and summarizing the latest reports of single-cell transcriptomics to define the novel promising targets for aptamer development.

Query 2: For figure 1, the label of RBD is wrong. RBD do not contain NTD.

2) Response:

We thank you the reviewer for the comment. We revised Figure 1 according to the reviewer’s suggestions. We apologize for the confusion and we specify in the figure legend the meaning of CTD and NTD.  

Query 3: The section 5, Single-cell Transcriptomics of SARS-CoV-2, seem unrelated to this paper, which should be deleted. Similar case is some part of section 2.1.

3) Response:

We thank you the reviewer for the comment. We revised paragraph 5 emphasizing that single-cell transcriptomics analysis might provide new insights to develop novel aptamers against novel receptors. As the review suggested, we believe that single-cell transcriptomics, Artificial Intelligence, and the novel receptors involved in SARS-CoV2 internalization are the uniqueness of this review. We would like to ask the reviewer to reconsider the revised version.  

Reviewer 2 Report

The review article of Paola amero et al. focuses on the use of aptamers against viruses, and the latest breakthroughs in terms of aptamers against spike protein and novel proteins involved in SARS-CoV-2 entry. The article is well-written, organized, and the logic of presentation is very clear.

In the context of aptamers against the HIV-1 gp120 component, the authors cannot overlook the G-quadruplex-forming aptamer ISIS 5320 capable of binding the V3 loop of the envelope glycoprotein gp120 and inhibit virus adsorption and cell fusion (Proc Natl Acad Sci U S A. 1994, 91, 1356-1360). Also, the studies of Hotoda and coworkers should be included, in which the 6-mer d(5′TGGGAG3′), known as “Hotoda’s sequence”, was identified as the lead sequence (J. Med. Chem. 1998, 41, 3655–3663), and later found to be active against HIV-1 at submicromolar concentrations only when conjugated at the 5′-position with bulky aromatic moieties (Bioconjugate Chem. 2007, 18, 1194–1204, Chem. Commun., 2010, 46, 8971–8973)Subsequent detailed studies on the 6-mer d(5′TGGGAG3′) better clarified the structure–activity relationships of G-quadruplex-forming oligonucleotides endowed with antiviral activity (Bioconjugate Chem. 2008, 19, 607–616, Chem. Commun. 2011, 47, 2363–2365, Chem. Commun. 2012, 48, 9516–9518).

Authors should add this part and relative references in the text.

Author Response

Query 1: The review article of Paola Amero et al. focuses on the use of aptamers against viruses, and the latest breakthroughs in terms of aptamers against spike protein and novel proteins involved in SARS-CoV-2 entry. The article is well-written, organized, and the logic of presentation is very clear.

In the context of aptamers against the HIV-1 gp120 component, the authors cannot overlook the G-quadruplex-forming aptamer ISIS 5320 capable of binding the V3 loop of the envelope glycoprotein gp120 and inhibit virus adsorption and cell fusion (Proc Natl Acad Sci U S A. 1994, 91, 1356-1360). Also, the studies of Hotoda and coworkers should be included, in which the 6-mer d(5′TGGGAG3′), known as “Hotoda’s sequence”, was identified as the lead sequence (J. Med. Chem. 1998, 41, 3655–3663), and later found to be active against HIV-1 at submicromolar concentrations only when conjugated at the 5′-position with bulky aromatic moieties 

(Bioconjugate Chem. 2007, 18, 1194–1204, Chem. Commun., 2010, 46, 8971–8973). Subsequent detailed studies on the 6-mer d(5′TGGGAG3′) better clarified the structure–activity relationships of G-quadruplex-

forming oligonucleotides endowed with antiviral activity (Bioconjugate Chem. 2008, 19, 607–616, Chem. Commun. 2011, 47, 2363–2365, Chem. Commun. 2012, 48, 9516–9518).

Authors should add this part and relative references in the text.

1) Response:

We thank you the reviewer for the positive comments and suggestions. We revised the manuscript according to the reviewer’s suggestions (please refer to lines 217-229). We also revised Table 2 according with the reviewer’s suggestions.

Reviewer 3 Report

In the main, this review has adequate documents for discussion about aptamer and its usage in SARS_Cov2 diagnosis. But it will be better that there is a comparison between primers & probes method or real-time and aptamer techniques. Furthermore, there are no evidences about Specificity, Sensitivity and limit of detection of Aptamer method in this review to aid reader for getting better conclusion.

Author Response

Query 1: In the main, this review has adequate documents for discussion about aptamer and its usage in SARS_Cov2 diagnosis. But it will be better that there is a comparison between primers & probes method or real-time and aptamer techniques. Furthermore, there are no evidences about Specificity, Sensitivity and limit of detection of Aptamer method in this review to aid reader for getting better conclusion.

1) Response:

We thank you the reviewer for the positive comments and suggestions. We revised the manuscript according to the reviewer’s suggestions (please refer to lines 382 to 419 and new table 3).